# Microstructural and Mechanical Characterization of Al Nanocomposites Using GCNs as a Reinforcement Fabricated by Induction Sintering

**DOI:** 10.3390/ijms24065558

**Published:** 2023-03-14

**Authors:** Verónica Gallegos Orozco, Audel Santos Beltrán, Miriam Santos Beltrán, Hansel Medrano Prieto, Carmen Gallegos Orozco, Ivanovich Estrada Guel, Roberto Martínez Sánchez, José Manuel Mendoza Duarte

**Affiliations:** 1Departamento de Nanotecnología, Universidad Tecnológica de Chihuahua Sur, Km. 3.5 Carr. Chihuahua a Aldama, Chihuahua 31313, Mexico; 2Departamento de Ciencias Básicas, Tecnológico Nacional de México, Campus Chihuahua II, Ave. de las Industrias #11101, Complejo Industrial Chihuahua, Chihuahua 31130, Mexico; 3Centro de Investigación en Materiales Avanzados (CIMAV), Laboratorio Nacional de Nanotecnología, Miguel de Cervantes No. 120, Chihuahua 31136, Mexico

**Keywords:** structural analyses, carbon nanostructures, mechanical properties, metal matrix composites, strengthening mechanism

## Abstract

High-energy ball milling is a process suitable for producing composite powders whose achieved microstructure can be controlled by the processing parameters. Through this technique, it is possible to obtain a homogeneous distribution of reinforced material into a ductile metal matrix. In this work, some Al/CGNs nanocomposites were fabricated through a high-energy ball mill to disperse nanostructured graphite reinforcements produced in situ in the Al matrix. To retain the dispersed CGNs in the Al matrix, avoiding the precipitation of the Al_4_C_3_ phase during sintering, the high-frequency induction sintering (HFIS) method was used, which allows rapid heating rates. For comparative purposes, samples in the green and sintered state processed in a conventional electric furnace (CFS) were used. Microhardness testing was used to evaluate the effectiveness of the reinforcement in samples under different processing conditions. Structural analyses were carried out through an X-ray diffractometer coupled with a convolutional multiple whole profile (CMWP) fitting program to determine the crystallite size and dislocation density; both strengthening contributions were calculated using the Langford–Cohen and Taylor equations. According to the results, the CGNs dispersed in the Al matrix played an important role in the reinforcement of the Al matrix, promoting the increase in the dislocation density during the milling process. The strengthening contribution of the dislocation density was ~50% of the total hardening value, while the contribution by dispersion of CGNs was ~22% in samples with 3 wt. % C and sintered by the HFIS method. Atomic force microscopy (AFM) and scanning electron microscopy (SEM) were used to analyze the morphology, size, and distribution of phases present in the Al matrix. From the analyses carried out in AFM (topography and phase images), the CGNs are located mainly around crystallites and present height profiles of 1.6 to 2 nm.

## 1. Introduction

The ball milling technique is a simple, economical, and high-performance manufacturing method used to fabricate metallic and multimetallic alloys, ceramic nanocomposites, etc. By mechanical milling, a homogeneous distribution of the reinforcement particles in the matrix is obtained. During the milling process, the metal matrix is plastically deformed while the grain size decreases at the nanometer scale, the dislocation density increases remarkably, and the reinforcing material becomes finer [1]. The next generation of nanomaterials is carbon-based, which includes carbon nanotubes, graphene, graphene oxide, fullerene, etc., which have been studied as reinforcing materials in metal matrix composites, due to their intrinsic physical and mechanical properties. Carbon allotropes such as carbon nanotubes, graphene, graphene oxide, fullerene, etc., are considered viable options to be used as reinforcements in the fabrication of aluminum matrix composites; these materials show increased physical and mechanical properties due to their peculiar physical and mechanical properties, low concentration required, and high reinforcement efficiency [2]. Carbon graphite nanostructures (CGNs) produced in situ by mechanical milling are viable economical options to produce metal matrix composites; the stress produced during the mechanical milling leads to the change in the multilayer graphitic structure to a few layers or even a monolayer [3]. One of the most important advantages of these materials is their relatively small volume fraction of the reinforcement concentration, which is required to obtain a significant increase in the mechanical properties. This aspect makes this type of composite a suitable material for industrial applications in aerospace engineering structures. Previous studies have shown that different graphite carbon nanostructures (GCNs) have been used as reinforcing materials, which can be produced when graphite is subjected to a high-energy ball mill [4]. Ramirez et al. use graphite-coated silver nanoparticles as a reinforcement in an Al matrix, increasing the elastic limit when the content of coated silver nanoparticles rises [5]. Hernandez and Calderon used graphite or fullerene (C+ soot) as reinforcing material to produce Al/Al_4_C_3_ nanocomposites, using SPS (spark plasma sintering). During the sintering process, the C nanostructures were transformed into the Al_4_C_3_ nanophase [6]. On the other hand, other authors (Yijun Liu et al.) report the manufacture of graphene/aluminum composites by in situ exfoliation of graphite through the friction stir method (FSP), and their results showed an increase in the microhardness values as a function of reinforcement content [7]. Bastwros et al. investigated the influence of graphene dispersion by mechanical milling and subsequent semisolid sintering of an Al6061 alloy on the mechanical properties and microstructure, obtaining significant hardening with graphene dispersion in the alloy matrix [8]. Mendoza et al. used Cu, Ni, and Ag to improve the interfacial bond between the aluminum matrix and the graphite nanoparticles in the fabrication of Al composites processed by mechanical milling, and their results showed that a low graphite content improves the mechanical properties of the Al matrix [3]. Mendoza et al. also prepared Al–graphite compounds using the HFIS method in order to retain the graphite particles in the microstructure of the compound after sintering, thus avoiding the precipitation of the graphite. Mendoza et al. also prepared Al-Graphite compounds using the HFIS method in order to retain the graphite particles in the microstructure of the compound after sintering, thus avoiding the precipitation of the Al_4_C_3_ phase [9]. In the HFIS method, the heating of green compact powder samples is conducted by high-frequency induction with the simultaneous application of pressure. One of the main advantages is that there is no physical contact between the heating coil and the sample, and this implies that there is lower contamination; additionally, the process is carried out at a relatively high heating speed, which ensures that the microstructure remains in a green state, with acceptable densification due to the high pressure applied [10]. In this work, the contribution of the effect of the dispersion of graphite nanostructures on the contributions to the hardening in the Al matrix, such as dislocation density, crystallite size, and particle dispersion, was evaluated. The structural analyses were carried out through an X-ray diffractometer coupled with a convolutional multiple whole profile (CMWP) fitting program to determine the crystallite size and dislocation density; both strengthening contributions were calculated using the Langford–Cohen and Taylor equations [11,12]. Nanocomposites were fabricated using high-energy mechanical milling and the application of the HFIS method for the sintering process. The obtained results were compared with samples in the green and sintered state through the CFS method.

## 2. Results

### 2.1. Microstructural Analysis

The microstructural analyses, phase morphology, and elemental content are presented through SEM-BSE (SEM-back scattered electrons) studies and EDS diagrams. The Al31-NS, Al31-S, and Al31-I samples are shown in Figure 1, Figure 2 and Figure 3, respectively. The dark and some bright particles of irregular shape are visibly well dispersed in the Al matrix of the Al-3NS sample, as can be seen from the image of Figure 1a. EDS elemental analysis was carried out on a zone with no evident presence of the second phase at this magnification (see selected area 1 of Figure 1b), which showed the presence of a significative C and Cu concentration (3.35 and 1.43 wt. %, respectively). A higher C quantity (5.2) was detected in a dark particle (see selected area 2 of Figure 1c), with the presence of Cu in a lower quantity (1.39). Elemental analyses in bright particles (selected areas 3 and 4 of Figure 1d,e) revealed the presence of Cu (7.2) with a remarkable presence of a relatively high C quantity around the bright particle (3.28).

Figure 2 shows the SEM-BSE image and EDS diagrams of the Al31-S sample. In the image, dark particles of irregular shapes and different sizes are observed, and the image also shows the presence of dark areas around some grains of ~0.5 μm in size. EDS elemental analysis carried out in a zone with the not-evident presence of the second phase at this magnification (see selected area 1 of Figure 2b) showed the presence of relatively low C and Cu (2.48 and 1.55 wt. %, respectively). Elemental analyses in the dark region around a grain showed about 1.72 of C and 2.03 of Cu (see selected area 2 of Figure 2c). Elemental analysis on a dark particle located in selected area 3 showed low C and Cu of approximately 1.8 and 1.83, respectively (see Figure 2d). The bright particle of selected area 4 showed a high Cu of 8.46 with 2.6 of C (see Figure 2e).

Figure 3 shows the SEM-BSE image and EDS diagrams of the Al31-I sample. The image shows dark particles of irregular shapes and sizes ranging from ~0.1μm to ~0.7 μm; the image also shows the presence of fine dark regions around some grains. EDS elemental analysis carried out in two dark particles of about 650 nm in size (see select areas 1 and 2 of Figure 3a–c) showed similar C contents of 2.34 and 2.29, respectively, and for Cu 2.05 and 1.83, respectively. Analyses in a zone with the not-evident presence of the second phase at this magnification (see selected area 3 of Figure 3a,d) showed the presence of low C and Cu (1.11 and 2.07, respectively). On the other hand, elemental analyses in the dark region around a grain showed a noticeable C content of about 3.6 and 1.46 of Cu (see selected area 4 of Figure 3a,e).

To further study the topography and phase morphology, AFM was used to study the Al31-NS, Al31-S, and Al31-I samples, and the corresponding images are shown in Figure 4, Figure 5, Figure 6, Figure 7, Figure 8 and Figure 9. Figure 4a–c shows the two-dimensional (2D) height topographic image, three-dimensional (3D) topographic image with a scan area of 500 nm × 500 nm, and typical phase AFM phase contrast images, respectively, for the Al31-NS sample. The images show crystallites that are mainly round and with a size in the range of 20 to 60 nm (see Figure 4a,b), while Figure 4c shows the presence of a second phase dispersed in the Al matrix (bright areas), located mainly at the crystallite boundary. Figure 5a,b show a higher-magnification image of a specific area of the phase contrast and height topographic image, respectively. Figure 5a shows the presence of barely distinguishable second-phase nanoparticles. The height profiles along the white line of three particles are shown in Figure 5c, and the location of the phase corresponding to each particle is indicated in Figure 5b (rounded zone). The height values of these particles are between 1.9 and 3.6 nm.

Figure 6a–c show the two-dimensional (2D) height topographic image, three-dimensional (3D) topographic image with a scan area of 457.7 nm × 500 nm, and typical phase AFM phase contrast images, respectively, for the Al31-S sample. The images show crystallite size in the range of 15 to 60 nm (see Figure 6a,b), while Figure 6c shows the presence of a second phase dispersed in the Al matrix (bright areas), located mainly at the crystallite boundary. Figure 7a,b show a higher-magnification image of a specific area of the phase contrast and height topographic image, respectively. Figure 7a shows the presence of barely distinguishable second-phase nanoparticles. The height profiles along the white line of three particles are shown in Figure 7c, and the location of the phase corresponding to each particle is indicated in Figure 7b (rounded zone). The height values of these particles are between 3.7 and 6.6 nm.

Figure 8a–c shows the two-dimensional (2D) height topographic image, three-dimensional (3D) topographic image with a scan area of 500 nm × 500 nm, and typical phase AFM phase contrast images, respectively, for the Al31-I sample. The images show crystallites that are mainly round and with a size in the range of 20 to 60 nm (see Figure 8a,b), while Figure 8c shows the presence of a second phase dispersed in the Al matrix (bright areas), located mainly at the crystallite boundary. Figure 9a,b show a higher-magnification image of a specific area of the phase contrast and height topographic image, respectively. Figure 9a shows the presence of barely distinguishable second-phase nanoparticles. The height profiles along the white line of three particles are shown in Figure 9c, and the location of the phase corresponding to each particle is indicated in Figure 9b (rounded zone). The height values of these particles are between 1.46 and 2.0 nm.

Figure 10a,b show the X-ray diffraction patterns for the not-sintered and sintered by CFS and HFIS conditions of the 3 wt. % C samples. The indexed diffraction patterns of Figure 10a show the characteristic Al peaks, and the X-ray diffraction profile magnification of Figure 10b is observed as an evident broadening peak due to grain refinement and the lattice distortion product of the high-energy milling process. On the other hand, DRX patterns also show the presence of the Al_4_C_3_ and Al_2_Cu phases for the composites sintered by the CFS method. As observed in Figure 10b, the detection of the Al_4_C_3_ phase is practically null for the sample sintered by the HFIS method. Table 1 summarizes the results of the dislocation density (ρ) and the mean values of the crystallite size (d) from the analysis of the X-ray diffraction peaks using the CMWP program. The highest values of dislocation density found correspond to the samples with 3 wt. % of C, while the smallest crystallite size was found in the Al31-NS sample.

### 2.2. Hardness Contribution Analysis

Establishing the microhardness (H) as the sum of each of the strengthening contributions, Equation (1) can be expressed as [13]:H = H_PN_ + H_SS_ + H_D_ + H_C_ + H_P_(1)
where H_PN_ is the Peierls–Nabarro strengthening hardness contribution, H_SS_ is the contribution caused by the solid solution, H_D_ is the dislocation contribution, H_C_ is the contribution by crystallite size, and H_P_ is the direct contribution by particle dispersion. The contribution to the hardness of the Peierls–Nabarro reinforcement, called lattice friction, is the reference resistance to dislocation and has a relatively low value [14]. The H_SS_ parameter is mainly related to the lattice parameter [15], and this lattice parameter varies slightly with composition and its impact is considered relatively low because of the low C and Cu content used in the nanocomposites. For practical purposes, the H_L_ parameter is considered as the sum of H_PN_ and H_SS_. The H_L_ parameter was calculated previously [16]: for pure Al samples this value is 29.48 VH and for the samples containing C and Cu the value is 25.3 HV. The strengthening hardness effect by dislocations, H_D_, is described by the modified Taylor Equation (2) [17,18]:H_D_ = kρ^1/2^(2)
where k = aMGb, G is the modulus of elasticity in shear, which is near to 26 GPa, b is Burger’s vector 0.2863 nm, a is the coefficient of the dislocation pattern hardness, M is the Taylor factor, and ρ is the dislocation density in the final condition. The strengthening contribution by the crystallite, H_C_, is described by Langford–Cohen Equation (3) [11,19]:H_C_ = k_1_d^−1^(3)
where d is the crystallite size and k_1_ = 6Gb.

Finally, the Orowan mechanism is based on the interaction of nanoparticles with dislocations; the effect of particle dispersion hardening is calculated by the following Equation (4):H_P_ = H_EXP_ – (H_L_ + H_C_ + H_D_)(4)

Table 2 shows separately the hardening contributions calculated in Vickers Hardness (VH) for each sample: H_L_, H_C_, H_D_, and H_P_. The table also includes the experimental microhardness, H_EXP_ (VH), and its corresponding standard deviation (SD).

Table 3 shows the microhardness values under different processing conditions, not sintered, CFS, and HFIS, for the pure Al sample and samples containing 0.75 and 3 wt. % C. For the non-sintered samples, the microhardness in Al31-NS samples increased by about 66% with the CGNs content compared to the Al-NS reference sample. After the sintering process using the CFS and HFIS methods, a decrease in the microhardness values was observed for all the samples. However, in the sample sintered by the HFIS method, the microhardness values remain above the samples sintered by the CFS method, as can be seen in Table 3.

On the other hand, the graph of Figure 11 shows each strengthening contribution of the Al matrix. Each term is added to the next and is represented graphically (H_L_, H_L_ + H_D_, H_L_ + H_D_ + H_C_, and H_L_ + H_D_ + H_C_ + H_P_, respectively), and the graph also includes the experimental microhardness curve (H_EXP_). From the graph, it is observed that there exists a close correlation between the experimental microhardness values and the sum of microhardness contributions (H_L_ + H_D_ + H_C_), which were calculated from XRD analyses for the Al-NS, Al-S, and Al-I samples. On the other hand, among all the contributions, the dislocation density provides an important contribution to the hardening of the Al matrix, and this value increases with the composition. For samples with low C content (0.75 wt. % of C), the contribution of H_D_ represents about 30% of the total microhardness value. On the other hand, for samples with high C content (3 wt. % of C), H_D_ represents about 50% of the total microhardness. Conversely, for the contribution by particle scattering for the samples with low C content, the H_P_ value represents approximately 30%, while for the samples with high C content the H_P_ value represents only 20%.

## 3. Discussion

During milling, the C works by (PCA) promoting the reduction in crystallite size, while the Cu works as an auxiliary element in the dispersion and integration of C (or nanostructured graphite), as previously reported in Al-C composites’ fabrication [20]. Numerous linear defects are induced, such as dislocations caused by the impact of milling media generating shear forces in the Al structure. As the milling continues, dislocations organize and form small-angle sub-boundaries, and upon further deformation results in the formation of fine crystallites [21]. The improvement of mechanical properties of the Al matrix is attributed mainly to a combination of some strengthening mechanisms, such as solid solution, dislocation, grain boundary, and Orowan effect [22]. Table 3 shows the microhardness values as a function of the composition and sintering condition (green state, CFS, and HFIS). A significant increase in the microhardness value is observed in the Al-NS sample of ~130 VH, compared to the value of 15 HV reported for pure Al [23]. The terms of contribution to the microhardness H_L_, H_C_, H_D_, and H_P_ are described in the graph of Figure 11, where the main contribution to the hardening in all the samples corresponds to the effect of the dislocation density (H_D_); for example, for the Al-NS sample the approximately 55% of the total hardening (H_EXP_) corresponds to the H_D_ term. The rest corresponds to the contributions of H_L_ and H_C_ (~22% and ~19%, respectively). The additional increase in microhardness observed in the samples containing C is mainly related to the dispersion of GCNs and the effect of the GCNs on the dislocation density generation. As observed in the graph of Figure 11, the contribution by particle dispersion, H_P_, does not contribute significantly to the total hardening (H_EXP_); for example, for the sample Al7525-NS, the H_P_ contribution found was of ~57.1 HV (35.5%), while that of the Al31-NS sample presents a value of ~43.3 HV (19.7%), see Table 2. However, a significant effect of the contribution of the dislocation density with the concentration of C was observed; that is, for the samples Al-NS and Al75/25-NS, the values of the H_D_ contribution were ~73.2 HV (55%) and ~54 HV (34%), respectively, while for the Al31-NS sample an important contribution of H_D_ was observed with a value of ~107.9 HV (49.4%), see Table 2. Therefore, it is assumed that the GCNs concentration dispersed in the Al matrix plays an important role in the reinforcing of the Al matrix, promoting the increase in dislocation density, with an increase of about 50% to the strengthening contribution of the Al matrix, in samples with a high content of C. The crystallite contribution to the Al matrix hardening, H_C_, did not have a significant effect on the GCN content; the average values for the Alp were ~24.8 HV, while for the samples with low content (75 wt. % of C) and samples with high C content (3 wt. % of C), the mean values were ~23.6 and ~41.9 HV, respectively.

After the sintering process by CFS and HFIS, the samples showed a decrease in the microhardness values, with a more pronounced decrease in the samples sintered through the CFS method. Long exposure times of the samples in the electric furnace result in an increase in the recovery, recrystallization, and crystallite growth processes, which can be retarded by a high heating rate through the use of the HFIS method. During the sintering process in the electric furnace, the nanostructured graphite is transformed into a fine phase of Al_4_C_3_. This phase was detected by XRD analysis performed on the Al31-S sample; the presence of this phase is observed in the enlarged image of the diffractogram of Figure 10b. The Al_4_C_3_ phase inhibits the excessive crystallite growth by the grain boundary pinning effect, and according to Zenner et al. [24], the pinning effect is promoted by reducing the size and increasing the volume fraction of the reinforcement. The grain boundary pinning effect was observed in the samples sintered by CFS, where the average crystallite size observed in the sample (Al-NS) before sintering was ~51.5 nm (See Table 1), while for the samples subjected to the sintering process (Al31-S and Al7525-S), the values found were ~68.7 nm and ~72.3 nm, respectively, slightly higher than those of the non-sintered sample. On the other hand, the relatively high contribution of the dislocation density, H_D_, observed in the Al31-S and Al7525-S samples after the CFS process (see Figure 11), is related to the thermal mismatch differences between the Al matrix and the Al_4_C_3_ phase, which causes an increase in the dislocation density, compensating for the excessive recovery process that occurs during the sintering process [25]. For example, the H_D_ contribution in the Al31-NS sample before sintering was approximately 107.9 HV, while after the sintering process, the H_D_ contribution value was 90.4 for the Al31-S samples. On the other hand, the HFIS sintered samples showed lower crystallite size and higher dislocation densities than the CFS sintered samples (see Table 2), which resulted in higher microhardness values (see Table 3). This result is attributed to the rapid sintering process that avoids the excessive recovery and recrystallization of the highly deformed Al matrix and the growth of the crystallite size. The high-speed sintering process also inhibited (to some extent) the transformation of the C phase to the Al_4_C_3_ phase, the XRD analysis performed on the Al31-I sample (see Figure 10b) detected a practically null presence of the Al_4_C_3_ phase (at 2 ϙ of 32° and 56°). Therefore, it is then assumed that in samples sintered by the HFIS method, the H_P_ contribution is due mainly to the dispersion of graphite nanostructures, with an increase of about 22% to the strengthening contribution of the Al matrix in samples with a high content of C. The SEM-BSE image analysis and the EDS elemental analysis performed on samples with 3% by weight of C under different processing conditions (green state, CFS, and HFIS) show the presence of mostly dark particles of different sizes and irregular shapes dispersed in the Al matrix. According to the elemental analysis, the particles contain mainly C and traces of Cu (see images in Figure 1, Figure 2 and Figure 3).

The dark areas correspond to the presence of nanostructured multilayer graphite; during the milling process, the graphite molecule exfoliates and adheres around the surface of the aluminum powder, which with subsequent mechanical milling this phase is located around the crystallites, which are formed in later stages of milling. Therefore, the irregularly shaped dark particles found in the SEM-EDS analyses correspond to grains or crystallites covered with a thin layer of nanostructured graphite. In addition, from the EDS elemental analyses the presence of Cu related to C particles was found, as the SEM image of Figure 1a shows a Cu particle coated with a thin layer of C (see selected areas 3 and 4). As previously reported, Cu was used as an auxiliary agent in the dispersion of CGNs in the Al matrix, improving the mechanical properties of the composites with increasing Cu concentration [26]. It is assumed that C adheres to the Cu surface during pre-milling during the preparation of C/Cu additive powder and is transported and integrated into the Al matrix through subsequent milling in the fabrication of the Al-C composites. Other authors have used Cu to improve the poor interfacial interaction between the graphite and aluminum matrix [27,28]. The presence of Cu/C was detected in different ratios in all regions (including visibly particle-free regions) analyzed by EDS in the 3 wt. % C samples (see Figure 1, Figure 2 and Figure 3), indicating a homogeneous dispersion in the matrix. The SEM-EDS analyses of the CFS-sintered Al3-S sample (see Figure 2a–e) shows some dark regions that are located mainly around the grain boundaries. The EDS analyses carried out in this zone (see selected zone 2) revealed an amount of 1.72 C, similar to that found in the particle from selected zone 3, with 1.8% of C, which indicates the presence of a second phase such as Al_4_C_3_. During the sintering process, C atoms diffuse towards the grain boundaries where they precipitate preferentially in these areas.

On the other hand, the Al31-I sample processed by the HFIS route presents dark C particles of irregular shape and of different sizes dispersed through the Al matrix (see Figure 3a), and with a microstructure similar to that found in the Al31-NS sample (see Figure 1a). Due to the short sintering process time, the C remains as a graphitic GCN. The AFM phase contrast mode image of the Al31-NS sample shows the distribution of a second phase around the crystallites (see Figure 4c and Figure 5b), this phase corresponds to the graphitic nanostructures found in the SEM and EDS analyses. Height profiles performed on some particles show heights between 1.9 and 3.6 nm. This thickness of the graphite nanostructures that cover the crystallites has been reported in surface coatings of Al with graphene sheets. Shengkai et al., using AFM measurements, estimated that monolayer graphene sheets had a thickness of 1.0 nm (although the theoretical value between graphene layers is 0.334 nm), considering measurement limitations at the nanometer scale of the AFM technique [29]. Similar results were observed in AFM images in the Al31-I sample analyzed in phase contrast mode (see Figure 8c and Figure 9b); in the image a second fine phase is observed around the crystallites, and the measurement of the height of three particles showed a maximum value of ~2 nm. From this, it can be deduced that in the samples sintered by HFIS and in the non-sintered state, the contribution of H_P_ corresponds mainly to the dispersion of graphite nanostructures. On the other hand, the AFM phase contrast mode image of the CFS-sintered Al31-S sample (see Figure 7a,b) shows the presence of nanoparticles around the crystallite whose morphology and particle size differ from those found in the crystal samples Al31-NS and Al31-I. The Al31-S sample particles are fiber-shaped, and from cross-particle height profile analysis they show a value between ~3.7 and ~6.6 nm. These particles correspond well to the Al_4_C_3_ phase found in the SEM and XRD analyses. It has been reported that theAl_4_C_3_ phase is detrimental for the corrosion resistance of the compounds, so several researchers have proposed different methods to integrate graphite nanostructures avoiding the formation of Al_4_C_3_. Table 4 shows a comparison between the mechanical properties of the Al–graphite nanostructures of this work and the results found in the literature of other authors.

## 4. Materials and Methods

The starting materials were Al powders (99.5% purity, mesh −325) and pre-milled graphite with Cu powder as an additive (C-Cu). Table 5 shows the sample compositions and the used nomenclature, low C (0.75 wt. %) and high C (3 wt. %) and their corresponding Cu content (0.25 and 1 wt. %, respectively). The table also includes pure Al as a reference. The additive (C-Cu) powders were milled for 4 h, and the Al-C-Cu powders for 8 h. The samples were mechanically processed in a high-energy Planetary mill (Metuchen, NJ, USA), and argon was used as the milling atmosphere. Used vials and milling media were made of hardened steel. The milling ball to powder weight ratio was kept at 5:1 (in wt.) and the sample weight was 5 g. For the additive (C-Cu), 1 mg of methanol was used as the process control agent and no process control agent was required for the Al-C-Cu composite fabrication (due to the lubricant properties of graphite). The milled powders were sintered using two routes: CFS and HFIS. Under the CFS method, samples were cold compacted under 1200 MPa pressure in a uniaxial load and then sintered at 550 °C at a heating rate of 20 °C/min for 1 h using under argon gas. Using the HFIS method, the samples were compacted and sintered (450 MPa at 450 °C) in a single step for 3 min, following a heating slope of 158 °C/min in an air atmosphere. Sample dimensions were 6 mm in height and 6 mm in diameter. The composites were studied by X-ray diffraction, scanning electron microscopy (SEM), and atomic force microscopy (AFM). The diffraction profiles were measured by a Philips X’pert powder diffractometer using a Cu cathode (l = 0.15406 nm). The step size and step time were 0.02° and 5 s, respectively. The X-ray diffraction peak profile analysis was carried out to determine the crystallite size distribution and the dislocation density of the nanocomposites studied using the CMWP fitting procedure program. Scanning electron microscopy images were acquired by a cold field emission JEOL JSM-7401 F microscope (JEOL LTD, Akishima, Tokyo, Japan) working at 5 and 17 kV to obtain images and elemental analysis, respectively. This SEM has an energy dispersive X-ray spectrometer (EDS) facility (Oxford Inca model, Oxford Instruments, High Wycombe, UK). Topography surface characterization was made using an atomic force microscopy tapping mode operated at 10 kHz to 1 MHz of the drive frequency range (Veeco Instruments, Inc., atomic force microscopy, Plainview, NY, USA). Images of compositional variations were analyzed using WSXM software [34] (WSxM v4.0 Beta 9.3 version, Nanotec Electrónica S.L., Centro Empresarial Euronova 3, Madrid, Spain) recorded from a phase angle difference between the excitation force and the tip response in amplitude modulation of AFM. The material hardness was measured by a Micro Hardness tester (Leco FM-07), using an indentation time of 10 s under a maximum load of 200 g.

## 5. Conclusions

Using high-energy ball milling, CGN-reinforced aluminum nanocomposites produced in situ were fabricated. In order to retain the CGNs dispersed in the Al matrix, avoiding the precipitation of the Al_4_C_3_ phase during sintering, the high-frequency induction sintering (HFIS) method was used (which allows high heating rates and therefore shorter processing times). The size of the crystallites and the contribution of the dislocation density to the strengthening were calculated from the microstructural analysis by using XRD. The CGNs dispersed in the Al matrix promote an increase in the dislocation density and a reduction in the size of the crystallites, which improves the mechanical response. The strengthening contribution of the dislocation density was ~50% of the total hardening value, while the contribution by dispersion of CGNs was ~22% in samples with 3 wt. % of C and sintered by the HFIS method. The dispersion of Al_4_C_3_particles contributed to the hardening of the Al matrix in sintered CFS samples, while the graphite nanostructures had an important effect on the hardening of samples in the non-sintered and HFIS-sintered states. From the analyses carried out in AMF (topography and phase images), the CGNs are located mainly around crystallites and present height profiles of 1.6.

## Figures and Tables

**Figure 1 ijms-24-05558-f001:**
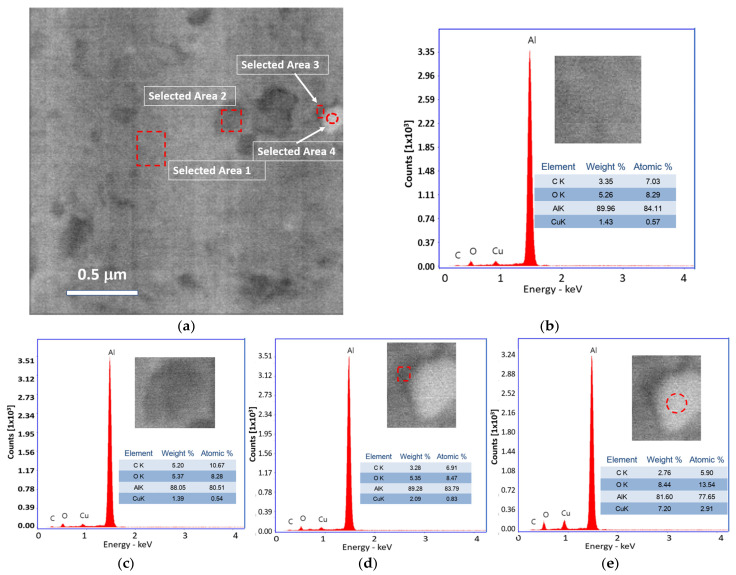
(**a**) SEM-BSE image of Al31-NS sample and EDS spectra. (**b**) Selected area 1, (**c**) selected area 2, (**d**) selected area 3, and (**e**) selected area 4.

**Figure 2 ijms-24-05558-f002:**
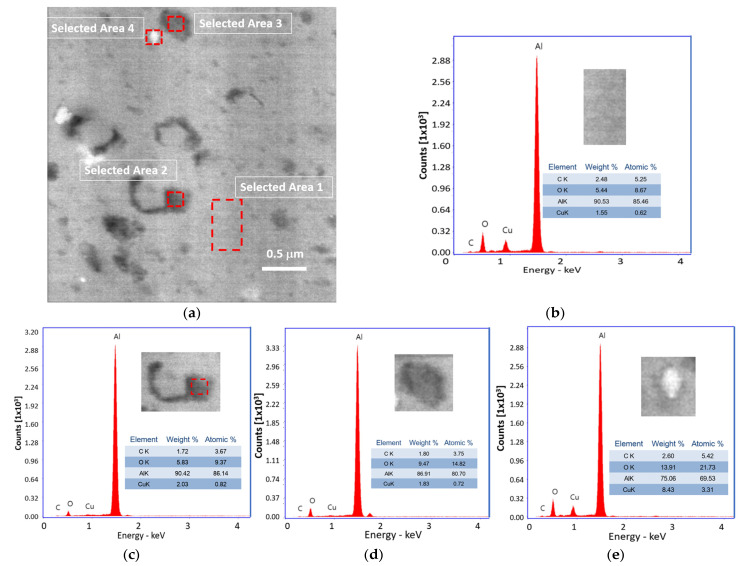
(**a**) SEM-BSE image of Al31-S sample and EDS diagrams for (**b**) selected area 1, (**c**) selected area 2, (**d**) selected area 3, (**e**) selected area 4.

**Figure 3 ijms-24-05558-f003:**
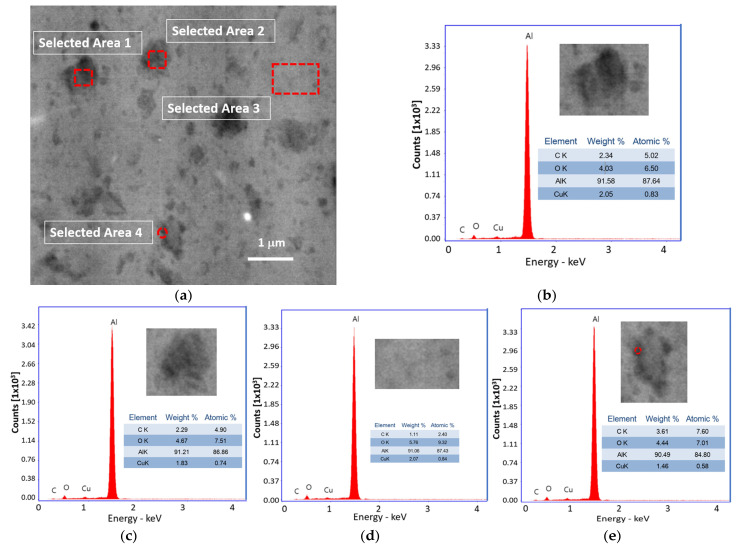
(**a**) SEM-BSE image of Al31-I sample and EDS diagrams for (**b**) selected area 1, (**c**) selected area 2, (**d**) selected area 3, (**e**) selected area 4.

**Figure 4 ijms-24-05558-f004:**
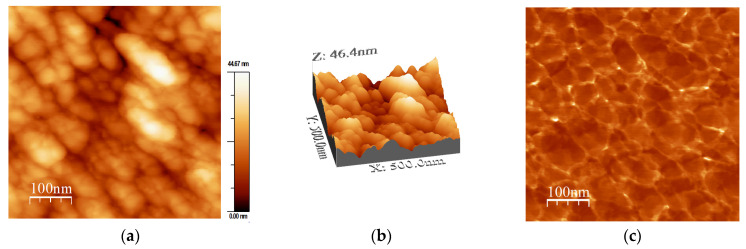
AFM images for Al31-NS sample: (**a**) 2D topography image, (**b**) 3D topographic images, and (**c**) phase morphology.4.

**Figure 5 ijms-24-05558-f005:**
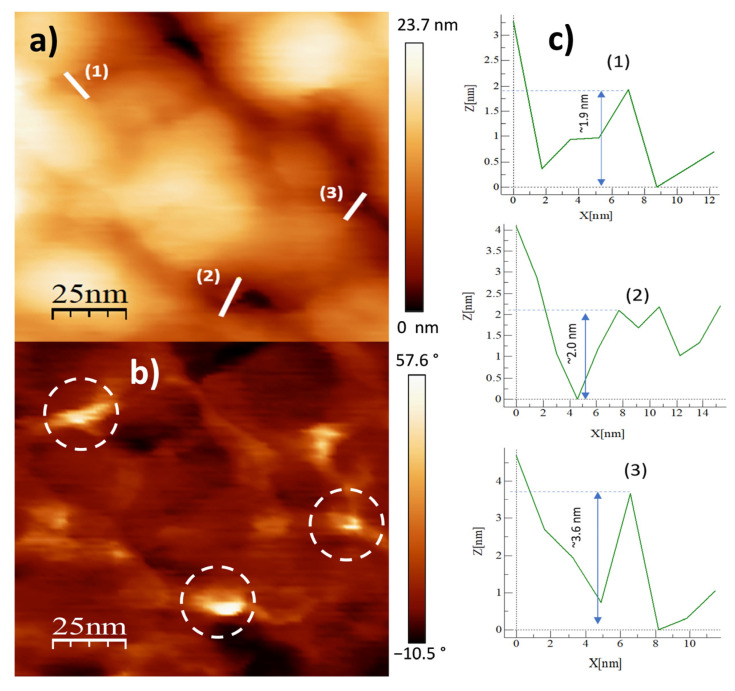
AFM images for Al31-NS sample: (**a**) topography image, (**b**) phase morphology, and (**c**) height profiles along the white line.

**Figure 6 ijms-24-05558-f006:**
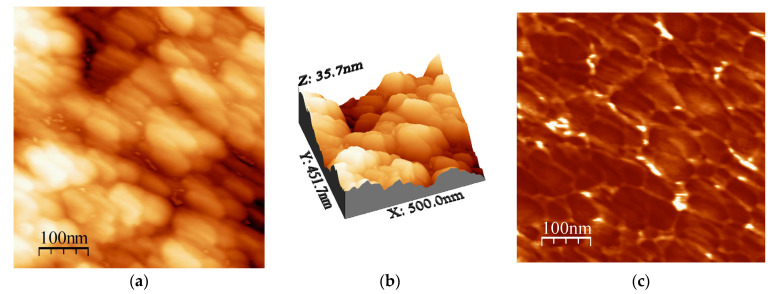
AFM images for Al31-S sample: (**a**) topography image, (**b**) 3D topographic image and (**c**) phase morphology.

**Figure 7 ijms-24-05558-f007:**
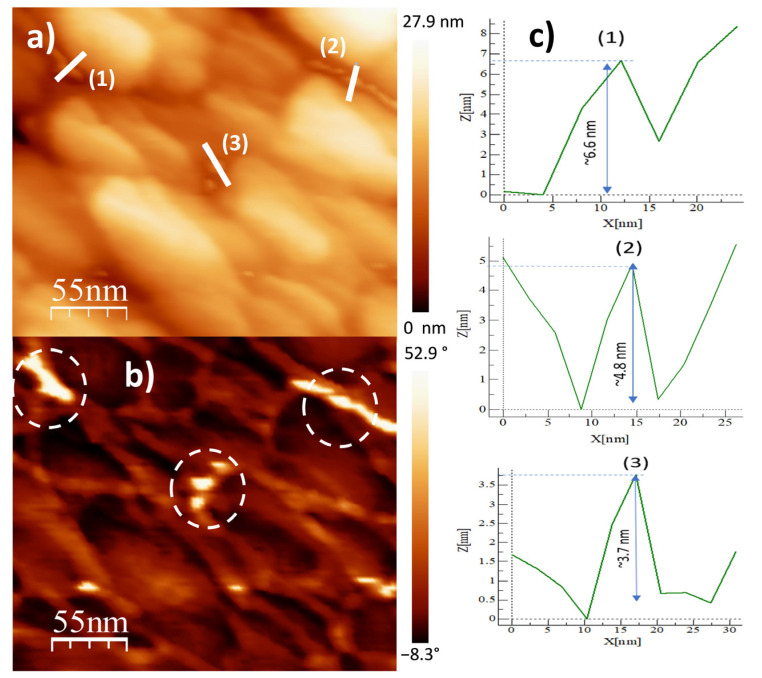
AFM images for Al31S sample: (**a**) topography image, (**b**) phase morphology, and (**c**) height profiles along the white line.

**Figure 8 ijms-24-05558-f008:**
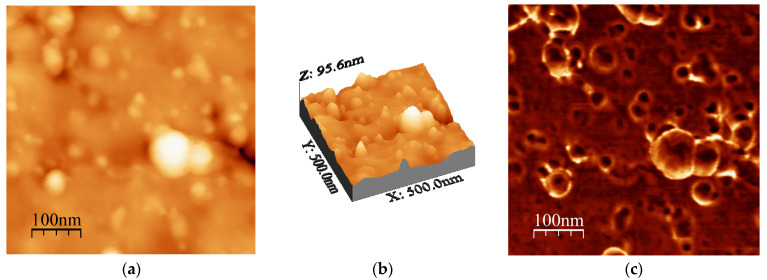
AFM images for Al31-S sample: (**a**) topography image and (**c**) phase morphology.

**Figure 9 ijms-24-05558-f009:**
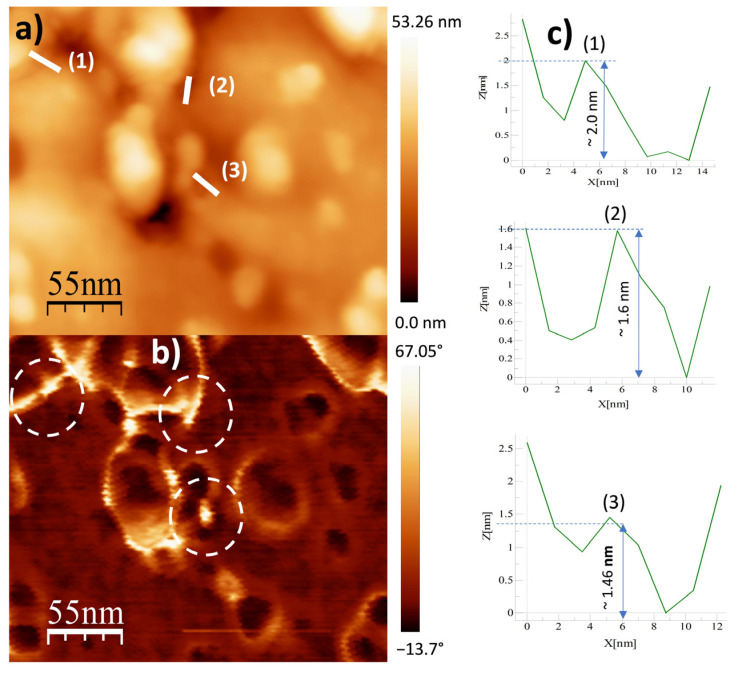
AFM images for Al31-S sample: (**a**) topography image and (**b**) phase morphology and (**c**) height profiles along the white line.

**Figure 10 ijms-24-05558-f010:**
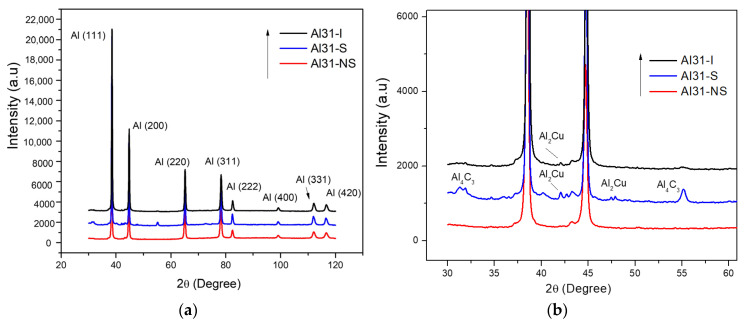
(**a**) X-ray diffraction spectrum for 3 wt. % C samples for not-sintered and sintered by CFS and HFIS conditions. (**b**) Image enlargement of X-ray diffraction profiles.

**Figure 11 ijms-24-05558-f011:**
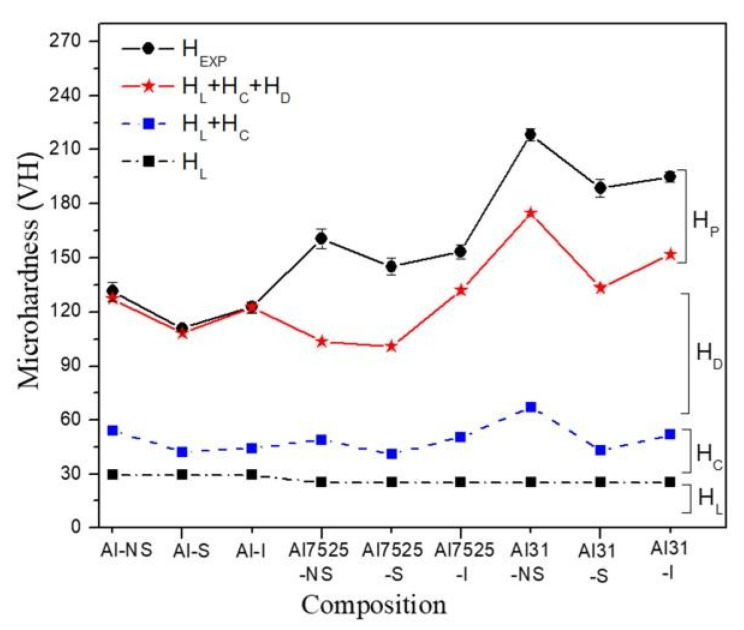
Contribution microhardness terms: H_L_, H_C_, H_D_, and H_P_ as a function of composition.

**Table 1 ijms-24-05558-t001:** Results from Rietveld and CMWP software.

Sample	*ρ* (×10^−14^ m^−2^)	*d* (nm)
Al-NS	24.6	51.5
Al-S	18.9	90.9
Al-I	27.3	82.6
Al7525-NS	12.5	51.9
Al7525-S	14.0	72.3
Al7525-I	24.3	49.4
Al31-NS	39.0	30.0
Al31-S	33.6	68.7
Al31-I	32.6	47.7

**Table 2 ijms-24-05558-t002:** H_L_, H_C_, H_D_, H_P_, and H_EXP_ values.

Sample	H_L_ (VH)	H_C_ (VH)	H_D_ (VH)	H_P_ H_EXP_ − (H_L_ + H_C_ + H_D_) (VH)	H_EXP_ (VH)	Std Dev.
Al-NS	29.48	24.8	73.2	4.03	131.54	5.22
Al-S	29.48	12.9	65.7	2.6	110.77	1.49
Al-I	29.48	14.7	78.4	–0.02	122.6	3.09
Al7525-NS	25.3	23.6	54.5	57.1	160.62	5.42
Al7525-S	25.3	15.7	59.7	44.3	145.11	4.64
Al7525-I	25.3	25.2	81.4	21.5	153.4	3.96
Al31-NS	25.3	41.9	107.9	43.3	218.44	3.24
Al31-S	25.3	17.7	90.4	55.3	188.83	4.79
Al31-I	25.3	26.54	100.1	43.2	195.16	2.85

**Table 3 ijms-24-05558-t003:** Microhardness as a function of composition and sintering condition (not sintered, CFS, and HFIS).

Sample	NS	Std Dev.	CFS	Std Dev.	HFIS	Std Dev.
Al	131.54	5.22	110.77	1.49	122.6	3.09
Al-75/25	160.62	5.42	145.11	4.64	153.4	3.96
Al-31	218.44	3.24	188.83	4.79	195.16	2.85

**Table 4 ijms-24-05558-t004:** Comparisons of mechanical properties of Al–graphite nanostructures with those from the literature.

Composition	Tensile Strength, σmax (MPa)	Microhardness (VH)	Method	Refs.
Al- 3 wt. %GNP/Cu	-	~195	Mechanical milling and sintered by HFIS	This work
Al- 0.5 wt. %graphene	~131	~50	Field-activated and pressure-assisted synthesis (FAPAS)	[30]
2024Al- 5 wt. %graphite	-	~117	Fraction stir processing (FSP)	[7]
Al- 23 wt. %multilayer graphite	~147	~48	Friction stir alloying (FSA)	[31]
1060Al- 1.5 wt. % graphene	497	165 *	Deformation-driven metallurgy (DDM)	[32]

* Calculated: microhardness (VH) ≈ σ**_max_**/3 [33].

**Table 5 ijms-24-05558-t005:** Compositions for studied Al-C/Cu and sample nomenclature (in wt. %).

Al	C	Cu	Not Sintered (Green)	Sintered (CSF)	Sintered (HFIS)
100.0	0.0	0.0	Al-NS	Al-S	Al-I
99.0	0.75	0.25	Al7525-NS	Al7525-S	Al7525-I
96.0	3.0	1.0	Al31-NS	Al31-S	Al31-I

## Data Availability

Data sharing is not applicable for this article.

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
