# Peer review of "Microstructural and Mechanical Characterization of Al Nanocomposites Using GCNs as a Reinforcement Fabricated by Induction Sintering"

_ijms, 2023, doi:10.3390/ijms24065558_

Round 1

Reviewer 1 Report

The authors used high-energy ball milling and high frequency induction sintering to fabricate CGNs-reinforced aluminum nanocomposites. The authors conducted microhardness testing, structural analysis, AFM and SEM to evaluate the properties and quality of nanocomposite.

The authors found that a low C concentration was needed to increased mechanical properties of the Al-C-Cu composites.

I have several questions need to be addressed before I can recommend the publication in the journal.

1. Is the 3% carbon concentration an optimal (or near optimal) value to improve the mechanical properties?

2. In line 322, "with low content (75 wt. % of C) and samples with high C content (3 wt. % of C)", is the (75 wt. % of C) sample a high or low C content sample?

Reviewer 2 Report

The article considers topical issues of creating and studying the properties of nanocomposites based on aluminum with carbon and copper additives. The problem of creating new composite materials has been studied very widely in recent years. Obtaining materials with new properties is necessary for the development of many innovative areas, especially in medicine, aircraft construction and space. Therefore, the article is relevant and interesting. The article has a number of comments and suggestions that need to be taken into account.

1. It is recommended to completely rewrite the abstract. Now abstract includes a short introduction and description of the work. The results are described very short and there is no actual data. Literature references are also usually not indicated in the abstract. The abstract should contain 1-2 sentences about the problem, as well as the results and conclusions.

2. Keywords are also recommended to expand.

3. Fig. 1 needs to be improved. Designate x, y axes. Remove the K and indicate 1 instead of 1.00, etc.

4. The same remark to figures 2 and 3. It would also be better to have the same y scale in all figures.

5. In figures 4-9, too much attention is paid to the visual representation of the obtained samples, while using different magnifications. It would be better to provide more quantitative information with composites characteristics. For example, graphs from these figures should be presented separately.

6. Fig. 11-a. Since different compositions of composites are presented, it is better to present them in the form of three columns for each parameter.

7. In section 3.3 it will be very useful to present a comparison table with the characteristics of the obtained samples and samples from other studies referred to by the authors. This will show the novelty of the results of this study and will make it possible to evaluate the advantages of new formulations in comparison with those obtained previously.

Round 2

Reviewer 2 Report

The authors corrected all the comments of the reviewer. The article has become better visualized and easier to understand. My opinion article can be recommended for publication.